# THE INTERPLAY BETWEEN DOMAIN SPECIALIZATION AND MODEL SIZE

## ABSTRACT

Scaling laws for language models have often focused on finding the optimal model size and token count for training from scratch. However, achieving this optimal balance requires significant compute resources due to the extensive data demands when training models from randomly-initialized weights. Continued pretraining offers a cost-effective alternative, leveraging the compute investment from pretrained models to incorporate new knowledge without requiring extensive new data. Recent findings suggest that data quality influences constants in scaling laws, thereby altering the optimal parameter-token allocation ratio. Building on this insight, we investigate the interplay between domain specialization and model size during continued pretraining under compute-constrained scenarios. Our goal is to identify an optimal training regime for this scenario and detect patterns in this interplay that can be generalized across different model sizes and domains. To compare general and specialized training, we filtered a web-based dataset to extract data from three domains: legal, medical, and accounting. We pretrained models with 1.5B, 3B, 7B, and 14B parameters on both the unfiltered and filtered datasets, then evaluated their performance on domain-specific exams. Results show that as model size increases, specialized models outperform general models while requiring less training compute. Additionally, their growing compute efficiency leads to reduced forgetting of previously learned knowledge.

## 1 INTRODUCTION

Recent advances in Language Models (LMs) have revealed emerging capabilities primarily attributed as a phenomenon of scale, achieved by training large models on large datasets using a causal language modeling objective (Brown et al., 2020; Ganguli et al., 2022; Srivastava et al., 2023; Wei et al., 2022). This paradigm aligns with scaling laws for Transformer LMs, which suggests that model size should scale proportionally with training tokens (Hoffmann et al., 2022). Hence, in scenarios with high availability of compute and data resources, training a larger model on all available data across multiple epochs becomes advantageous.

However, given the high costs associated with training LMs (Dubey et al., 2024; Touvron et al., 2023; Brown et al., 2020; Wei et al., 2022), there is a growing demand for techniques that add new knowledge into these models without the need for complete retraining. Continued pretraining leverages existing knowledge of pretrained LMs through transfer learning, providing a more compute-efficient alternative to training from randomly-initialized weights. This process employs techniques such as compute-equivalent replay, learning rate re-warming, and re-decaying to minimize the forgetting of past knowledge (Ibrahim et al., 2024).

When LMs are applied within a particular domain, continued pretraining on domain-specific data becomes advantageous (Rozière et al., 2024). This approach enhances the model's performance within the targeted domain while potentially degrading it in general contexts, given the forgetting of past knowledge (Junior et al., 2024). Despite this trade-off, significant performance improvements have been observed in specialized models on domains such as medicine (Chen et al., 2023; Labrak et al., 2024), law (Colombo et al., 2024b; Chalkidis et al., 2020; Polo et al., 2021), and programming (Rozière et al., 2024; Li et al., 2022; 2023).

Thus, LM development is not confined to scenarios with abundant computing resources, required for training general-purpose models like GPT-4 (OpenAI et al., 2024). In compute-constrained

scenarios, optimizing the selection of training data becomes crucial to achieve better performance with less compute. This raises a question: when training resources are limited, is it beneficial to filter the data for domain specialization, starting from a pretrained general model?

In this study, we explore the interplay between model size and domain specialization to uncover evidence addressing this question. Our goal is to identify trends that indicate the optimal training regime. We hypothesize that larger models, with more trainable parameters, exhibit greater capacity to retain learned knowledge. Therefore, as model size increases, the benefits of specialization diminish, given that a larger model would have the capacity to learn both specialized and general knowledge.

To test this hypothesis, we applied neural topic filters to a web-based dataset, creating three subsets focused on legal, medical, and accounting data. We then trained models with 1.5B, 3B, 7B, and 14B parameters on both the filtered subsets and the original unfiltered dataset. Their performance was evaluated using domain-specific standardized multiple-choice exams.

Contrary to our initial hypothesis, when trained with fixed compute resources (one epoch on the unfiltered dataset), the specialized models outperformed their general counterparts with reduced computation across all model sizes. As shown in Figure 1(a), the results reveal a power-law: in a compute-constrained scenario, specialized models achieve better performance than general models, maintaining this relation as model size increases. Additionally, specialized models achieve their minimum perplexity with a decreasing number of training steps, as shown in Figure 1(b). In particular, the 14B specialized model achieves lower perplexity while using $4.3\times$ less compute than its general counterpart.

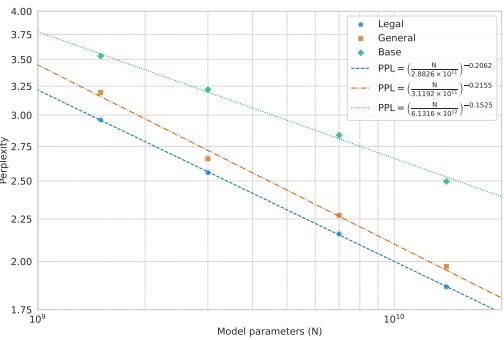

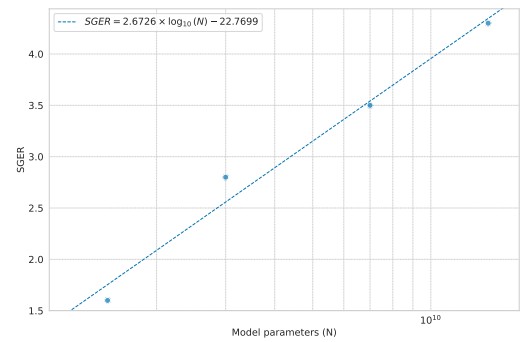

(a) Perplexity vs. model size on a legal test suite.

(b) Specialized-to-General Efficiency Ratio (SGER) (defined in Equation 3) vs. model parameters.

Figure 1: Each point represents the checkpoint with the lowest perplexity in a held-out legal test suite for either legal, general, or base models. In **(a)**, a power-law relationship is observed: as model size increases, specialized models consistently outperform general models. In **(b)**, the compute-effectiveness gap between specialized and general models is quantified using the SGER metric, showing that as model size increases specialized models achieve their lowest perplexity with less training steps.

The main contribution of this work lies in providing, to the best of our knowledge, the first evidence, grounded in observed power-law trends, that domain specialization in Transformer LMs yields increased performance and compute efficiency over general training, as model size increases under a compute-constrained scenario. Contrary to the common assumption that a sufficiently large model can learn both domain-specific and general knowledge without drawbacks, our results show that, as model size increases, specialized models outperform general models in their target domain, with increasing sample-efficiency. Furthermore, specialized models exhibit reduced forgetting of previously learned knowledge, benefiting from this increased sample-efficiency as model size increases.

## 2 RELATED WORK

Recent studies have demonstrated favorable results for continued pretraining, particularly with domain specialization, highlighting it as a cost-efficient alternative across various domains by leverag-

ing previously acquired knowledge. Although we do not formalize a scaling law in this work, our observations are grounded in established literature, which suggests variations in scaling laws due to slight changes in data characteristics and training scenarios.

## 2.1 CONTINUED PRETRAINING ON DOMAIN-SPECIFIC DATA

The CodeLlama models (Rozière et al., 2024) were developed by continually training Llama 2 (Touvron et al., 2023) on code specific data. It was observed that continued pretraining on code data outperforms training models with the same architecture from scratch, as shown by evaluations on popular code-to-description benchmarks. Notably, the Code Llama 7B model, trained on 500 billion code tokens, matched the performance of the Llama 2 70B model, which was trained on 2 trillion unique tokens of general content. The Code Llama series also outperformed domain-specific models trained from scratch, such as AlphaCode (Li et al., 2022) and StarCode (Li et al., 2023). These findings highlight the benefits of leveraging pre-existing knowledge from a base model to enhance specialized training, demonstrating scalability from 7B to 70B parameters.

In the legal domain, models such as SaulLM-7B (Colombo et al., 2024b), LegalBERT (Chalkidis et al., 2020), and Bertikal (Polo et al., 2021) have pioneered continued pretraining on legal data. In (Colombo et al., 2024b), the Mistral-7B model (Jiang et al., 2023) was continually pretrained on 30 billion legal tokens, resulting in SaulLM-7B. By applying techniques to mitigate forgetting, the authors achieved superior performance on legal benchmarks compared to the base model. In a follow-up study (Colombo et al., 2024a), the authors scaled up the experiments, training SaulLM-54B and SaulLM-141B on 540 billion legal tokens. Whereas scaling generally improved the performance of Mixture-of-Experts (MoE) models, certain tasks exhibited a negative correlation with scale, leading to degraded performance on general-context tasks relative to the base models.

The literature suggests a trend in which continued pretraining on domain-specific data scales equally in terms of training data and model parameters. However, in scenarios with compute constraints, it remains unclear whether, as model size increases, general training eventually outperforms domain-specialized training. In other words, does domain specialization lose its effectiveness for sufficiently large models, which could potentially learn both general and domain-specific knowledge without drawbacks? In our study, we address this question through the proposed experiments.

## 2.2 SCALING LAWS

The work by Kaplan et al. (2020) proposes a power-law relationship between performance, training data, and model parameters. By extending the analysis from Kaplan et al. (2020) with a broader range of models, data, and hyperparameter tuning, Hoffmann et al. (2022) found that model size and token quantity scale together to achieve compute optimal training. The Chinchilla-70B, trained on 1.4 trillion tokens, outperforms models with similar compute budgets but less data, supporting their scaling hypothesis. This proposed power-law remains widely accepted in the literature, supporting subsequent studies that explore scaling laws for LMs in different scenarios, including our own.

Hernandez et al. (2021) explore transfer learning under data-constrained regimes through domain specialization with code data, starting from a model pretrained on text. They propose a power-law for data transfer, demonstrating that pretraining offers compute benefits by leveraging prior knowledge acquired during the pretraining process. However, with less data constraint, transfer learning can hinder performance due to the ossification of prior knowledge. While the work by Hernandez et al. (2021) shares certain limitations with Kaplan et al. (2020), such as the lack of learning rate tuning for each token budget (Hoffmann et al., 2022), the authors observed how different training regimes can lead to changes on classic scaling laws. Our study explores a similar scenario, focusing on the interplay between domain-specific and general training as model size increases, rather than limiting the analysis to transfer learning within a single domain.

Scaling laws for Transformer LMs (Hoffmann et al., 2022) have proven replicable across multiple studies (DeepSeek-AI et al., 2024; Muennighoff et al., 2023; Gu et al., 2024; Ye et al., 2024; Que et al., 2024). Variations in training regimes can influence these scaling laws, as observed in data-constrained settings (Muennighoff et al., 2023) and scenarios involving high-quality data (DeepSeek-AI et al., 2024). Our study focuses on domain-specific continued pretraining. Existing studies propose power-laws that help determine the compute optimal mixture ratio between general

and domain-specific datasets for this type of pretraining (Gu et al., 2024; Ye et al., 2024; Que et al., 2024). These power-laws offer guidance on optimal mixture ratios for specific regimes, such as data-constrained domain-specific continued pretraining, given a target general or domain-specific document reconstruction loss (Que et al., 2024).

In contrast to prior studies on domain-specific continued pretraining, which focus on varying the mixture-ratio between general and domain-specific data to predict the reconstruction loss on domain-specific documents (Gu et al., 2024; Ye et al., 2024; Que et al., 2024), our analysis focus on domain specialization from a different perspective. We curate a domain-specific subset from within a single general dataset, then train models exclusively on either this filtered subset or the full, unfiltered dataset. Unlike prior works, we keep the mixture-ratio constant, comparing full domain specialization directly against the entire unfiltered dataset. This approach allows us to isolate the effect of domain specialization as model scale increases. Hence, we ask whether scaling alone is sufficient to negate, or instead amplify, the benefits of full specialization. To the best of our knowledge, this is the first study to empirically address this question, providing power-laws for the interplay between model size and domain specialization in downstream task performance.

## 3 METHODOLOGY

We perform continued pretraining on both specialized and general datasets, then evaluate the resulting models across three benchmarks: a new specialized domain, a new general domain, and one that more closely resembles the original pretraining domain. In this section, we present the datasets used for model training, the evaluation data, and the experimental setup.

### 3.1 PRETRAINING DATA

Our experiments compare two training regimes: general and specialized. For general training, we use a web corpus without domain-specific curation. For specialized training, we filter this corpus using neural topic classifiers to select content from specific domains.

The pretraining data for the proposed experiments was derived from ClueWeb 2022 (Overwijk et al., 2022) dataset, a web corpus sourced from a commercial web search engine. For the initial preprocessing, we extracted text from HTML web pages and applied the filter proposed by Rae et al. (2022), which removes non-natural language content, such as tables, long lists, and texts with excessive symbol count. Given the multilingual nature of the dataset, we sourced only web pages predominantly in Portuguese, as identified by the dataset's language tag. This choice was motivated by Portuguese's underrepresentation in existing open datasets for training LMs. Consequently, we expected a pronounced effect of specialization in our continued pretraining experiments.

Recent studies (Abdin et al., 2024; 01.AI et al., 2024; Dubey et al., 2024) show a trend towards using LMs to filter pretraining data based on specific topics or subjective criteria, such as "quality" or text "coherence/cohesion", depending on the bias of the larger LM (Soldaini et al., 2024). Following this methodology, we trained three classifiers to identify legal, medical, and accounting content. We selected these three domains due to their high availability of standardized exams administered multiple times per year, which enables us to evaluate recently released models while reducing concerns about training-test set contamination.

To extract domain-specific content from a general dataset, we developed filters to classify web pages with domain-specific content, following the methodology used in FineWeb (Almeida et al., 2025). We first employed gpt-4o-2024-08-06 to assess domain-specific relevance in a small sample of the preprocessed ClueWeb 2022, using a scale from 0 to 5, where 0 indicates no domain-specific content and 5 indicates highly domain-specific content. Using these labeled examples, we then finetuned a Portuguese version of BERT (Souza et al., 2020) to classify the entire dataset, enabling large scale classification across billions of tokens. The results of this filtering process, as well as further details regarding the training of the neural topic classifiers, are presented in Appendix C.

We applied three filters to the preprocessed ClueWeb 2022 dataset, creating three domain-specific datasets for our experiments. Our analysis revealed that ClueWeb 2022 has a high legal data distribution, with a high concentration of web pages receiving high scores. Leveraging this, we extracted 8.4 billion tokens of legal data by selecting all web pages with scores above 3. In contrast, ClueWeb

2022 had a low amount of medical and accounting data, with most web pages scoring between 0 and 1. To obtain a large enough pretraining dataset for those domains, we use lower filtering thresholds of 1.4 and 1.0 for the medical and accounting domains, respectively, which yielded 8.4 billion tokens for both domains, matching the legal dataset's size. Additionally, we created a fourth dataset comprising all preprocessed ClueWeb 2022 web pages without domain filtering, totaling 58 billion tokens. The specialized datasets are used to train domain-specific models, while the general dataset serves to train models with broad knowledge.

## 3.2 EVALUATION

Typically, works addressing the scaling laws of Transformer LMs use the perplexity metric to measure performance on document reconstruction tasks from a development dataset partition, examining how model size relates to performance gains (Kaplan et al., 2020; Gu et al., 2024; Hernandez et al., 2021; Hoffmann et al., 2022). These datasets may introduce noise into the evaluation process, as they are randomly sampled from web pages. To mitigate this issue, we use document collections specifically designed to assess human knowledge, which may better reflect the real learning of domain-specific information. Thus, in all experiments, we assess model performance using Multiple-Choice Question Answering (MCQA) on standardized human exams.

As discussed in Xia et al. (2023), discontinuous metrics such as accuracy on correct answers can obscure emerging trends as we scale compute, data or parameters. To address this, we follow their suggested evaluation approach, which assesses model performance on MCQA benchmarks by calculating the perplexity of the token representing the correct alternative letter:

$$\text{Perplexity} = 2^{-\log_2 P(\text{correct}|\text{prompt},\text{question},\text{choices})} \tag{1}$$

where $P(\text{correct}|\text{prompt}, \text{question}, \text{choices})$ is the probability assigned by the model to the correct answer token (A, B, C, D or E) given the few-shot prompt (with three examples in our experiments), the question, and the set of four or five choices (depending on the exam). We report the overall score for each model as the average perplexity for the correct letter across the total amount of questions for a test suite.

Table 1 presents the statistics for all MCQA evaluation datasets used in this study. We consider five test suites: three domain-specific evaluations for medical, legal, and accounting evaluations in Portuguese; a new general knowledge evaluation using recent Portuguese exams; and an original general knowledge evaluation in English, closely aligned with our base models' pretraining dataset. Each dataset consists of standardized multiple-choice exams designed for humans. The number of answer choices ranges from four to five, with only one correct response. For questions requiring image understanding, we either removed them or, when available, replaced the image with its caption.

Table 1: Statistics of all benchmarks.

| Benchmark | Questions | Publication date | Test suite |
|---|---|---|---|
| OAB 2023 | 240 | 02/26/2023 | Legal |
| OAB 2024 | 238 | 03/24/2024 | Legal |
| REVALIDA 2023 | 181 | 03/05/2023 | Medical |
| REVALIDA 2024 | 187 | 07/20/2024 | Medical |
| CFC 2023 | 100 | 05/07/2023 | Accounting |
| CFC 2024 | 99 | 06/30/2024 | Accounting |
| CPNU 2024 | 370 | 08/18/2024 | New general |
| BNDES 2024 | 455 | 08/13/2024 | New general |
| ENADE 2023 | 702 | 11/26/2023 | New general |
| MMLU College | 828 | 09/07/2020 | Original general |
| MMLU High school | 3860 | 09/07/2020 | Original general |

We evaluate models' ability to solve the Bar Association Exams (OAB), designed to assess knowledge of the Brazilian legal domain in humans. The OAB is the mandatory bar exam required for anyone wishing to practice law in Brazil, serving as an entry level assessment of legal knowledge. For our legal test suite, we use OAB exams from 2023 and 2024, totaling 478 questions.

To evaluate medical knowledge, we use the REVALIDA exam, which assesses foreign-trained doctors' understanding of medical practice in Brazil. This exam is the primary requirement for revalidating foreign medical diplomas. Our medical test suite consists of REVALIDA exams from 2023 to 2024, totaling 368 questions.

The Federal Accounting Council's (CFC) Proficiency Exam is designed to assess humans' knowledge of the rules and skills required to practice as an accountant in Brazil. To obtain official registration as an accountant in the country, a graduate must pass this exam. Our accounting test suite comprises CFC exams from 2023 and 2024, totaling 299 questions.

Our neural topic classifiers may primarily be selecting higher-quality data. In this case, the improvements observed in our specialized training may not stem from true domain specialization but rather from training on curated data. To investigate this hypothesis, we evaluate our trained models on CPNU, BNDES and ENADE exams that assess human knowledge across various domains, excluding those that require legal, medical, and accounting knowledge. These exams, taken in 2023 and 2024, form our new general test suite. Since they involve Portuguese across multiple disciplines, continued pretraining on general Portuguese data may enhance model performance on these questions. However, this improvement must surpass the gains observed in domain-specific Portuguese training.

In continued pretraining, the forgetting of previously learned knowledge occurs naturally as models adapt to new data. Since specialization datasets are smaller and less diverse, focusing on a specific domain, does specialization increase forgetting? To investigate this, we evaluated the models using the MMLU benchmark, which assesses general knowledge in English (the primary domain of Qwen2.5's training). We selected the College and High School partition as our original general test suite, comprising 4,688 questions.

### 3.3 Experimental setup

We extended the pretraining of Qwen2.5 (Qwen et al., 2024), our base model, at scales of 1.5, 3, 7, and 14 billion parameters to examine how model size influences domain specialization, assessing both the benefits and limitations of domain-specific training. We selected this model family for its multilingual capabilities and availability in multiple pretrained sizes, focusing on the four smallest versions for our study, released on September 19, 2024.

The models were trained using the causal language modeling objective and the AdaFactor optimizer (Shazeer & Stern, 2018). We used a learning rate of 0.001 and a warm-up of 250 steps. The batch size was set to 512, with a sequence length of 4,096 tokens. Each model was trained for a total of 58.7 billion tokens (28,000 steps), equivalent to one epoch of our unfiltered (general) training dataset. In total, we trained four models for the legal, medical, accounting, and general dataset, amounting to 16 models over 28,000 steps. Since the specialized datasets are a smaller, filtered subset of the general dataset, specialized models were trained over seven epochs, while general models were limited to one epoch.

The primary goal of this study is to identify which training regime, specialized or general, optimally leverages available data at a compute-constrained scenario. To simulate this constrained scenario, we fixed the total compute budget for all experiments to match the cost of one epoch on the unfiltered dataset. Despite differences in the number of unique tokens, both the specialized and general training regimes were allocated the same compute resources, ensuring a fair basis for comparison.

To calculate training compute for each experiment, we used this approximation from Kaplan et al. (2020):

$$C \approx 6ND \tag{2}$$

such that $C$ represents the compute, in FLOPs, used for training the model, $N$ is the number of trainable parameters, and $D$ is the number of tokens seen during training.

To quantify the compute-effectiveness gap between specialized and general models, we propose the metric SGER given by:

$$\text{SGER}(N) = \frac{C_g(N)}{C_s(N)} = \frac{6ND_g(N)}{6ND_s(N)} = \frac{D_g(N)}{D_s(N)} \tag{3}$$

where $C_g(N)$ represents the compute spent to achieve the minimum perplexity for a general model of size $N$, using the same approximation from Equation 2, while $C_s(N)$ refers to the equivalent value for a specialized model. Similarly, $D_g(N)$ and $D_s(N)$ represent the amount of tokens used from the general and specialized datasets for a model of size $N$ to achieve the minimum perplexity, respectively.

Our analysis focuses solely on the minimum perplexity points from the respective domain-specific test suites for all plots in this work. Consequently, even when evaluating models on the new general knowledge benchmarks in Portuguese and English, we use the checkpoint that achieved the lowest perplexity in the corresponding domain-specific test suite. This experimental setup targets the intended compute-constrained, domain-specific scenario, revealing power-laws that indicate particularities of each training regime in this scenario.

## 4 RESULTS AND DISCUSSION

In this section, we discuss the main findings of this study, focusing on the legal domain, which our domain classifier has identified as having the highest volume of high-quality data in the source collection. The results for each checkpoint of every model across all test suites are provided in Appendix B.

Analysis of the filtered datasets revealed a significantly higher distribution of legal data in ClueWeb 2022. Consequently, for legal training, we selected data with a higher likelihood of being genuinely legal. In contrast, for the medical and accounting domains, training the topic classifiers proved challenging due to the lower distribution of relevant data.

### 4.1 SPECIALIZED MODELS ACHIEVE LOWER PERPLEXITY ON TARGET DOMAIN

Each specialized model exhibits lower perplexity than its general counterpart, as shown in Figure 1(a), 2(a), and 2(b). While no clear trend indicates that the perplexity gap increases with model size for the legal model, Figure 2 suggests an increase in this gap for the medical and accounting models as the model size grows.

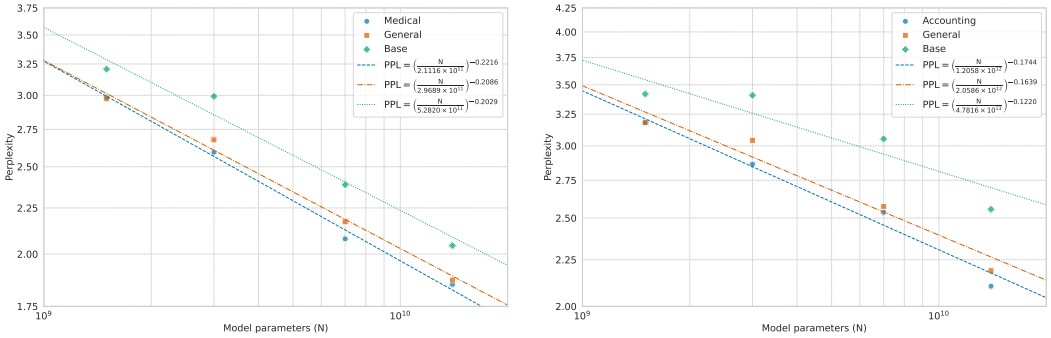

(a) Perplexity vs. model size on the medical test suite.  (b) Perplexity vs. model size on the accounting test suite.

Figure 2: Each point represents the checkpoint with the lowest perplexity on a held-out, domain-specific test suite for medical, accounting, general, or base models.

Given the potential limitations of medical and accounting specialization due to the small sample of domain-specific data in the general dataset, we hypothesize that, with a larger sample, domain specialization would behave similarly to the legal model, showing neither an increase nor a decrease in

the perplexity gap. However, it is worth noting that even in domains with limited data, specialization still yields improvements, as observed in the medical and accounting models.

## 4.2 Specialized models present increasing sample-efficiency

As shown in Figure 1(b), the SGER increases from $1.6\times$ at 1.5B parameters to $4.3\times$ at 14B parameters. These results indicate that, as the number of trainable parameters grows, specialized models require fewer tokens to reach their minimum perplexity, increasing its sample-efficiency. For example, the 1.5B specialized model requires 33.6B tokens, whereas the 14B model needs only 12.6B tokens.

This trend suggests that increasing model size actually enhances the compute benefits of specialization. While larger models may have a greater capacity to retain the full knowledge of an unfiltered dataset, they show superior capacity for specialization when trained on the filtered version of the dataset. This suggests that specialization improves the sample-efficiency of larger models in compute-restricted scenarios, underscoring the advantages of data filtering for domain specialization.

This increased sample-efficiency and the inherently smaller dataset required for specialization may introduce potential drawbacks during training. As shown in Figure 3, the optimal model size for a given amount of compute increases more steeply with specialization than with general training. This implies that larger models should be used for specialized training to better leverage available compute resources. However, training larger models introduces engineering challenges and leads to higher serving costs compared to smaller ones.

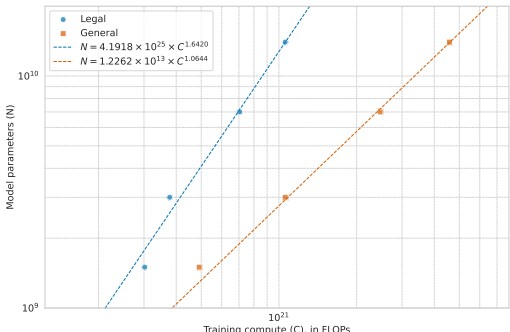

Figure 3: Optimal model size for a given amount of available compute, comparing legal and general models on the legal test suite.

## 4.3 Specialized models present diminishing forgetting

In this experiment, we hypothesize that the original general knowledge test suite primarily assesses the core knowledge acquired during Qwen2.5's initial pretraining, as it covers a broad range of domains in English, possibly, the predominant language used in training these models. As shown in Figure 4, we observe that as model size increases, specialized models achieve lower perplexity compared to the general ones. This improvement appears to be a byproduct of the specialized models' higher sample-efficiency. With fewer training steps required to reach their minimum, these models experience less forgetting of previously learned knowledge.

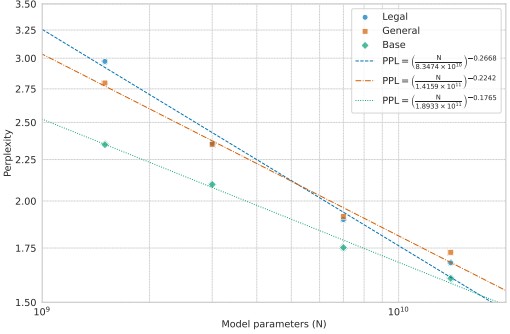

Figure 4: Perplexity on the original general knowledge test suite vs. model size.

### 4.4 SPECIALIZED MODELS EXHIBIT HIGHER PERPLEXITY ON GENERAL DOMAIN

We suspected that the topic filter might be selecting only higher-quality data. As a result, the better performance observed in the specialized models could be attributed only to data curation, rather than domain specialization. To test this hypothesis, we evaluated both general and specialized models on our new general knowledge test suite in Portuguese. If the performance was solely due to curation of quality data, the specialized model should achieve lower perplexity than the general model in these scenarios.

However, as shown in Figure 5, we observed that models trained on the complete general dataset achieved better perplexity than the specialized models. Based on this, we conclude that we are indeed studying a real scenario of domain specialization.

Supported by the findings in Que et al. (2024), we postulate that expanding the training dataset while maintaining the general to domain-specific data ratio would sustain all these observed trends. However, in a high data regime, the emphasis shifts from the applicability of domain-specific training to the challenges of continued pretraining, such as forgetting (Ibrahim et al., 2024), where previously learned knowledge is lost.

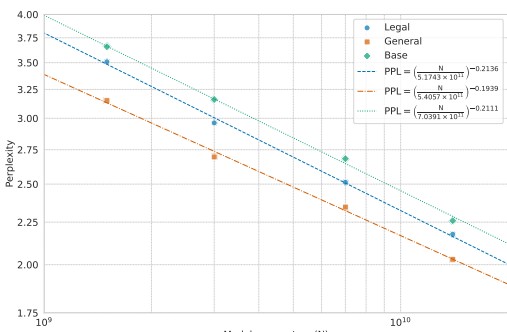

Figure 5: Perplexity on the new general knowledge test suite vs. model size.

## 5 CONCLUSION

In this study, we explored the interplay between model size and domain specialization during continued pretraining in a compute-constrained scenario, aiming to identify trends that reveal the optimal training regime. To the best of our knowledge, this is the first study to provide evidence, based on observed power-law trends, that domain specialization in Transformer LMs yields improvements in both performance and compute efficiency as model size increases, when compared to general training.

Evaluation on domain-specific MCQA exams revealed a power-law: as model size increases, the performance gap between specialized and general models in the target domain persists, with specialized models achieving superior results at larger scales across all three tested domains. Although these findings are based on a small number of instances and are limited by the natural distribution of domain-specific content in our training dataset, they provide evidence that specialized training can outperform general training in compute-constrained scenarios.

Additionally, results in legal benchmarks further highlight power-laws in specialization. As model size increases, specialized training becomes more sample-efficient, achieving lower perplexity with less training steps. However, this also suggests that, given a fixed compute budget, the model size required to reach an optimal perplexity in the specialized regime tends to increase compared to general training. Consequently, larger specialized models exhibit lower forgetting rates, as they require fewer continued pretraining steps to reach minimum perplexity.

Future studies could formalize scaling laws specific to the interplay explored in our work, providing additional evidence for the trends we identified. To achieve this, our methodology could be replicated across model sizes, spanning more orders of magnitude, to strengthen the evidence base for fitting a power-law. Additionally, extending training to incorporate datasets from diverse sources would help determine whether this trend persists across different dataset mixtures and domains.

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

## A LIMITATIONS

We aim to generalize the trends identified in our experiments by finding the most efficient training approach between domain specialization and general continued pretraining. However, our experiments were limited to a single dataset, and potential noise may exist due to inaccuracies in the content classifiers used to create the specialized datasets. Thus, we cannot yet confirm that this trend is broadly applicable, as we are restricted to our conclusions on the ClueWeb 2022 dataset.

Our conclusions in this study are based on four data points for each power-law in both the specialized and general models. Although we observed a good fit at scales from 1.5B to 14B parameters, ideally, we would have conducted experiments using larger ranges. However, we are constrained by the compute costs of training larger models. Using smaller models is also not an option, as models with fewer than 1.5B parameters do not perform better than chance, likely due to the difficulty of our domain-specific benchmarks.

Finally, our study faces the same validity challenges as other works evaluating LMs in specialized domains. What constitutes medical, legal, or accounting knowledge? Does a high LM score on a standardized MCQA exam correlate with superior performance on other tasks within the same domain? By focusing on the specialization regime, our study evaluates the learning of knowledge classified as legal, medical, and accounting through the topic classifier's understanding of these topics. Additionally, in attempting to capture this implicit understanding, we risk a potential sampling bias in our selection of evaluation benchmarks, which we classify as capable of measuring this domain-specific knowledge.

## B TRAINING RESULTS BY MODEL

Figures 6, 7, and 8 present the results for the legal, medical, accounting, and general models against three test suites: target domain, new general, and original general. Checkpoints were saved every 2,000 steps for 28,000 steps (58.7B tokens). Each figure is organized in three rows: the top row presents results on the domain-specific test suite, the middle row on the new general test suite, and the bottom row on the original general test suite.

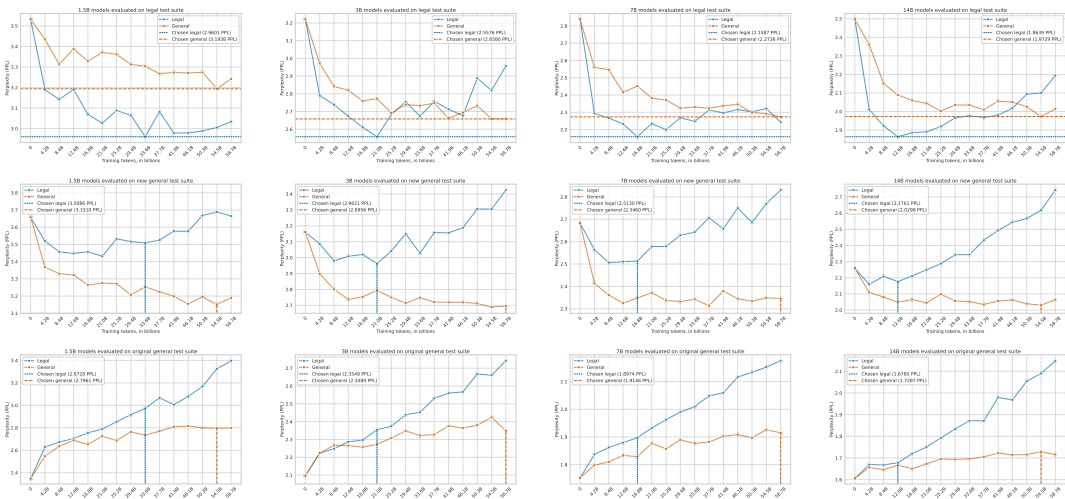

Figure 6: Evaluation of legal models on target domain (top row), new general (middle row), and original general (bottom row) test suites.

The points labeled as "Chosen" correspond to the checkpoints selected for power-law analysis. As described in Section 3.3, for each training run, the checkpoint with the lowest perplexity on the domain-specific test suite was selected. These checkpoints are marked by horizontal lines in the top row of Figures 6, 7, and 8, and are also highlighted in the second and third rows through vertical lines.

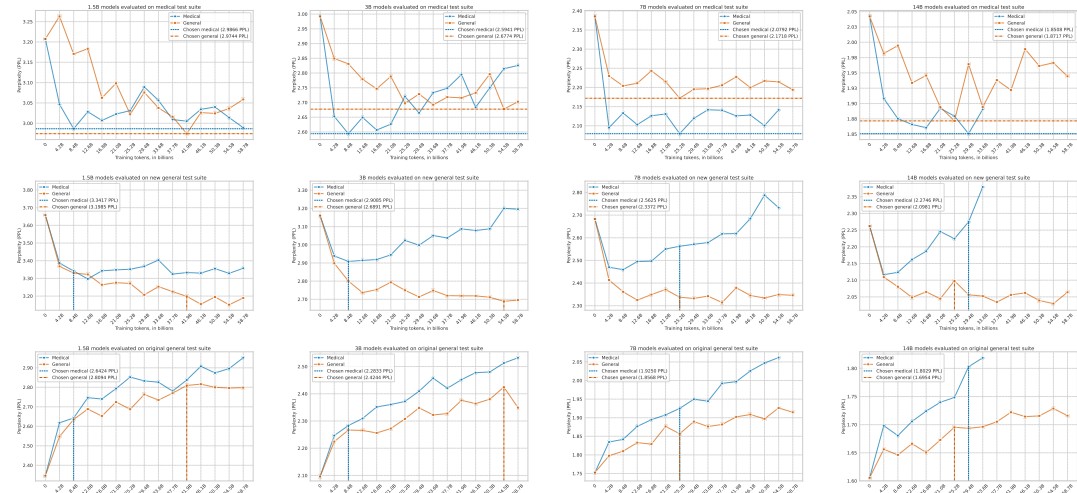

Figure 7: Evaluation of medical models on target domain (top row), new general (middle row), and original general (bottom row) test suites.

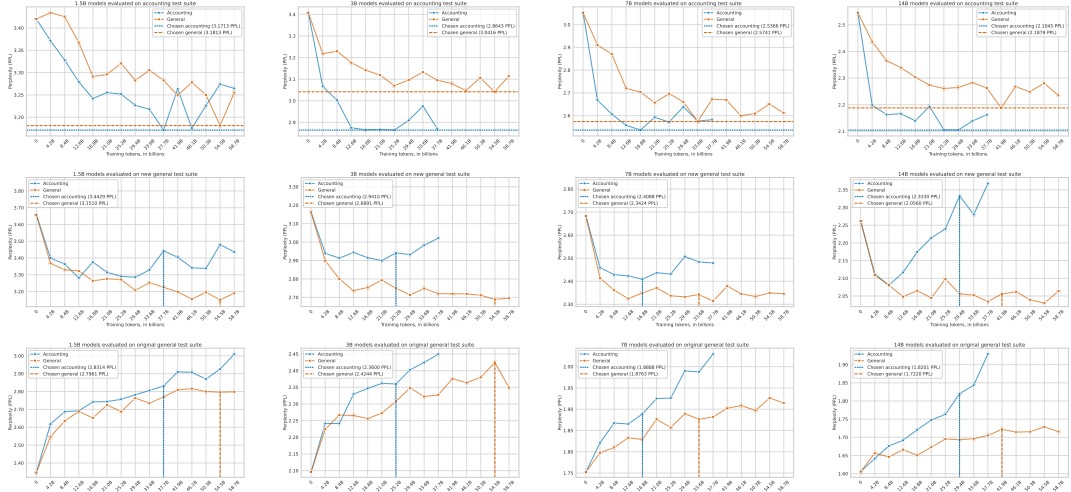

Figure 8: Evaluation of accounting models on target domain (top row), new general (middle row), and original general (bottom row) test suites.

As shown in the ClueWeb 2022 domain-specific relevance score distribution (Figure 9), the dataset contains a low amount of medical and accounting relevant content, with lower domain specificity scores for these areas compared to the legal domain. This may explain the absence of a power-law trend in the results of the medical and accounting models on the new and original general test suites. This hypothesis is further supported by the narrower performance gap observed in the first row plots of Figures 7 and 8 compared to Figure 6, suggesting that the medical and accounting models retained less domain-specific knowledge during training.

Additionally, Figures 7 and 8 show missing points for the domain-specific models. Early experiments with the legal models revealed that, due to their smaller dataset size, performance tends to saturate earlier compared to the general models. Beyond this saturation point, no further improvements are observed and perplexity begins to increase. To avoid unnecessary spent compute, for the medical and accounting models, training was stopped early once this point was reached.

## C  NEURAL TOPIC CLASSIFIERS

A sample of documents from ClueWeb2022 was annotated using the gpt-4o-2024-08-06 model to assess their relevance to the medical, accounting, and legal domains. The annotation employed a scoring scale ranging from 0 (no domain-specific content) to 5 (highly domain-specific content), based on the prompts presented in Figures 10, 11, and 12. These labeled examples served to distill the ability of a larger model to assign domain-specific relevance scores, enabling a smaller model to replicate this task more efficiently at inference time.

Initial experiments revealed a scarcity of relevant documents in the medical and accounting domains, with a high prevalence of label 0 (non domain-specific content). To balance the label distribution for classifier training, additional samples were collected. The final distill dataset comprised 120,000 labeled examples for the legal domain, 130,000 for the medical domain, and 150,000 for the accounting domain.

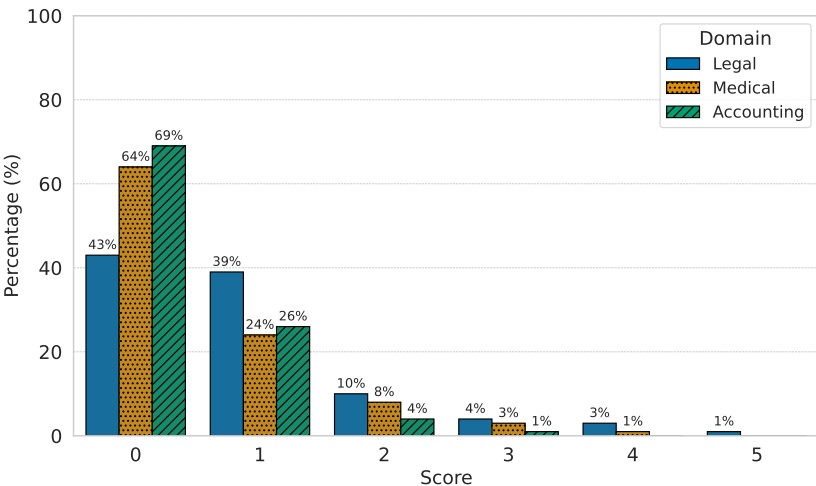

Figure 9: ClueWeb 2022 domain-specific relevance score distribution across three specialized domains.

For the distillation process, the Portuguese pretrained BERT model BERTimbau was selected due to its superior performance in early experiments. A classification head was added to predict one of five relevance classes per document, being the only set of weights updated during training. The model was trained for 20 epochs with a 5% warmup phase, using the AdamW optimizer, 0.0003 for learning rate, and a cosine decay schedule. This process produced three neural topic classifiers, one for each target domain.

Due to BERT's limited context windows and the high inference cost of these LMs, classification was performed using only the first 512 tokens of each document. The resulting distribution of domain-specific relevance scores for the medical, accounting, and legal domains is presented in Figure 9.

Below is a passage from a web page. Evaluate whether the page contains Brazilian legal knowledge using the 5-point scoring system described below. Points are accumulated based on the fulfillment of each criterion:

- Add 1 point if the passage contains legal terms. This includes pages that may contain advertisements or promotional material but still include relevant legal terms.
- Add another point if the passage cites any legal norms, such as laws, rulings, or regulations. The presence of such citations indicates some level of legal content, even if mixed with non-legal material or presented in a disorganized manner.
- Award a third point if the passage contains information about the Brazilian legal context. This may include simple explanations or introductions to legal concepts, even if not well-organized.
- Award a fourth point if the passage presents the content of a legal norm. It must be coherent and focused on the Brazilian legal context.
- Award a fifth point if the passage is highly informative, well-organized, and provides deep insights into legal topics. It should be free from irrelevant content and offer a thorough understanding of Brazilian legal issues.

Passage: <passage>

After examining the passage:
- Briefly justify your total score in up to 100 words.
- Conclude with the score using the format: "Legal Score: <total points>".

Figure 10: Prompt used for assessing legal relevance.

Below is a passage from a web page. Evaluate whether the page contains Brazilian medical knowledge using the 5-point scoring system described below. Points are accumulated based on the fulfillment of each criterion:

- Add 1 point if the passage contains medical terms. This includes pages that may contain advertisements or promotional material but still include relevant medical terms.
- Add another point if the passage cites any medical norms, such as guidelines, protocols, reports, or drug manuals. The presence of such citations indicates some level of medical content, even if mixed with non-medical material or presented in a disorganized manner.
- Award a third point if the passage contains information about the Brazilian medical context. This may include simple explanations or introductions to medical concepts, even if not comprehensive or well-organized.
- Award a fourth point if the passage presents the content of medical norms. It must be coherent and focused on the Brazilian medical context.
- Award a fifth point if the passage is highly informative, well-organized, and provides deep insights into medical topics. It should be free from irrelevant content and offer a thorough understanding of Brazilian medical issues.

Passage: <passage>

After reviewing the passage:
- Briefly justify your total score in up to 100 words.
- Conclude with the score using the format: "Medical score: <total points>".

Figure 11: Prompt used for assessing medical relevance.

Below is a passage from a web page. Evaluate whether the page contains Brazilian accounting knowledge using the 5-point scoring system described below. Points are accumulated based on the fulfillment of each criterion:

- Add 1 point if the passage contains accounting terms. This includes pages that may contain advertisements or promotional material but still include relevant accounting terms.
- Add another point if the passage cites any accounting norms, such as tax legislation, resolutions, technical manuals, or accounting pronouncements. The presence of such citations indicates some level of accounting content, even if mixed with non-accounting material or presented in a disorganized manner.
- Award a third point if the passage contains information about the Brazilian accounting context. This may include simple explanations or introductions to accounting concepts, even if not comprehensive or well-organized.
- Award a fourth point if the passage presents the content of accounting norms. It must be coherent and focused on the Brazilian accounting context.
- Award a fifth point if the passage is highly informative, well-organized, and provides deep insights into accounting topics. It should be free from irrelevant content and offer a thorough understanding of Brazilian accounting issues.

Passage: <passage>

After reviewing the passage:
- Briefly justify your total score in up to 100 words.
- Conclude with the score using the format: "Accounting score: <total points>".

Figure 12: Prompt used for assessing accounting relevance.

