# OpenReview forum: "The interplay between domain specialization and model size"
_ICLR.cc/2026/Conference — Submitted to ICLR 2026_

### Official Review · Reviewer_AxPP · 2025-10-31

**Soundness:** 1
**Presentation:** 2
**Contribution:** 2
**Rating:** 2
**Confidence:** 4

**Summary:**

This paper aims to explore the patterns in domain specialization (i.e., continuing pretraining on corpora in a specific domain). The authors propose the following hypothesis:
“Domain specialization in Transformer LMs yields increased performance and compute efficiency over general training, as model size increases under a compute-constrained scenario”.
To verify this hypothesis, the authors conduct continued pre-training on LLMs with different sizes from 1.5B to 14B on general-domain corpora and domain-specific corpora (a subset of the general-domain corpus extracted by a classifier) to acquire several “general models” and “specified models”.
Experimental results show that (1) Specialized model exhibits lower perplexity than general models on domain-specific test sets; (2) Specialized models achieve minimum PPL with fewer tokens than general models, and this difference on the number of tokens required increases with model size; and (3) Specialized models are prone to have less forgetting on general knowledge as they are further trained on fewer tokens.

**Strengths:**

(1)  This paper aims to explore compute-optimal continued pretraining, a significant research direction.

**Weaknesses:**

(1)  Writing is needed to improve, as it is difficult to understand:
a)	The key concept of this paper, “domain specialization”, is not well explained in this paper. The biggest confusion I had when reading this article was in which aspects a good "domain specification" should be quantified? Is it by comparing the lowest ppl achieved under the same computational budget (i.e., 6ND)?
b)	In line 60 the authors first propose a hypothesis “larger models exhibit greater capacity to retain learned knowledge”, while later in line 68 they deny their own hypothesis, and in line 97 they make another claim “domain specialization in LLMs yields increased performance and compute efficiency over general training as model size increases”. It took me a long time to understand that the real claim this paper is making is the last one.
c)	In section 4.1 the authors list two contradictory conclusions: “the gap” increases with the model size in medical and accounting domains, while it does not increase in the legal domain, without further exploration. I also don’t quite get how the “gap increase” conclusion is reached: though in figure 2 the slopes of the fitted blue and orange lines are slightly different, there doesn't seem to be such a pattern if you only look at the four data points.
(2)  The evaluation data size is too small to draw convincing conclusions. The domain-specific evaluation data only contains 199, 368, and 478 data points. As a result, all experiment results are based solely on the PPL of a few hundred tokens (1 token per data point). To verify a universally applicable scaling law of LLMs you should at least add some experiments with larger evaluation data and in high-resource languages.
(3)  Some of the results are already well-known in the community. For example, (1) training on domain-specific corpora usually yields lower PPL on the corresponding domain than mixing additional unrelated text, and (2) larger models usually need fewer tokens to reach the same performance (in “Scaling Laws for Neural Language Models“ paper). Although experiment (2) was conducted by training from scratch, intuitively this conclusion can also easily be extended to continued training, especially if the data distribution of pre-training data and continued training differs significantly (mainly English+Chinese v.s. Portuguese).
(4)  I wonder the applicability of the conclusions in this paper. For instance, what advice can this paper provide if I want to train a specialized model for the legal domain within a given FLOPS budget? Should I prioritize scaling the model parameters, the amount of training data, or put more effort into data cleaning? I think this is a more important question in continued pretraining at a compute-constrained scenario.

**Questions:**

Firstly, refer to weakness 2 and 4.
In addition, could you add some figures with curves showing how the test loss changes as training progresses, similar to Figure 2 in the "Scaling Laws for Neural Language Models" paper? This would be beneficial for displaying more information to intuitively compare the differences between various models and data.

---

> ### Author Response · Authors · 2025-11-20
> **Response to Reviewer AxPP**
>
> Thank you for taking the time to review our work.
>
> Below, we focus on addressing the weaknesses raised and clarifying some points where we believe there are **misunderstandings about what we actually did and claimed.**
>
> > **(2) The evaluation data size is too small to draw convincing conclusions…**
>
> We follow the evaluation methodology from **[1]** to evaluate MCQA downstream performance: the perplexity is computed on the token corresponding to the correct alternative letter (A/B/C/D/E), conditioned on the full few-shot prompt, question text, and answer choices. **This does not mean we only use "one token per data point".**
>
> While many scaling law works **[2, 3, 4]** use the reconstruction error on web text, we deliberately choose MCQA exams as our evaluation, aiming for a metric more aligned with the possible downstream performance of the model. In this setting, the PPL of the correct answer choice is more suitable for scaling analysis than accuracy due to its continuous nature, as noted in **[1]**. **This choice is, to our knowledge, unique to our scope and requires restricting evaluation to exams outside the training cutoff date to mitigate contamination, which naturally limits the available data.**
>
> Finally, we do not claim a "universally applicable scaling law of LLMs"; we explicitly state that we do not formalize a new scaling law, but rather observe power-law trends in a controlled continued pretraining scenario (Qwen2.5, Portuguese domain specialization, fixed compute). **Given this scope and level of claims, we consider that our methodology is adequate.**
>
> > **(3) Some of the results are already well-known in the community…**
>
> You highlight three premises as "already well-known": (1) training on domain-specific corpora tends to yield lower PPL on that domain, (2) larger models need fewer tokens to reach a given performance, and (3) these two assumptions are true even under a transfer learning scenario.
>
> **However, our contribution is not to restate these premises (as detailed in Sections 1 and 2, mainly on lines 96 to 103 and 166 to 176), but to study their "interplay" under a controlled continued pretraining setup:** same base family (Qwen2.5), same "source" corpus, fixed FLOPs, and a direct comparison between domain specialization (topic-filtered subset) and general training (unfiltered) via power-law analysis. By studying the interplay between both training regimes in this controlled scenario, we present the main contributions of this study:
>
> - Specialized LMs consistently outperform their general counterparts on the target domain, and this advantage remains stable as the number of trainable parameters increases (Figure 1a);
>
> - Specialized LMs exhibit better and increasing sample efficiency, with the relative compute efficiency benefit of specialization itself following a log-linear relationship with model size (e.g., SGER rising from 1.6x at 1.5B to 4.3x at 14B in the legal domain, Figure 1b);
>
> - Specialized LMs show decreasing forgetting of prior learned knowledge when compared to general continued pretrained models, as model size increases (e.g., on MMLU, Figure 4).
>
> **These findings are not immediate corollaries of the "already well-known" premises you mention:** they emerge from the analysis of the interplay between the specialized vs. general power-laws, rather than from analyzing each curve in isolation. Moreover, our conclusions are grounded in downstream exam performance per domain, which is another distinctive aspect of our study.
>
> > **(4) I wonder the applicability of the conclusions in this paper…**
>
> **In this work, we do not study how data quality or cleaning interacts with the scaling behavior of domain specialization versus general training.**
>
> Within the scope of our results, one can conclude that, given a fixed compute budget, it is beneficial to perform topic filtering on the corpus and then use the power-laws in Figure 3 to select the compute-optimal model size, i.e., the model size that achieves the lowest perplexity on the relevant test suites for that compute, as remarked in Section 4.2.
>
> Regarding the requested plots of test loss as training progresses, these are reported in Section B of our supplementary material (Figures 6, 7, and 8), which show the behavior of all models on all test suites throughout training.
>
> **In summary, we disagree with the weaknesses raised, as we believe they largely stem from misunderstandings of our experimental design and evaluation protocol.** Most importantly, we would welcome references to prior work that directly supports the view that our contributions rely on "well-known" assumptions in the scenario we study, for further discussion.
>
> ---
>
> ## References
>
> **[1]** Training trajectories of language models across scales.
>
> **[2]** Scaling laws for neural language models.
>
> **[3]** Scaling laws for transfer
>
> **[4]** Training Compute-Optimal Large Language Models

---

### Official Review · Reviewer_Tb5q · 2025-11-01

**Soundness:** 3
**Presentation:** 3
**Contribution:** 3
**Rating:** 6
**Confidence:** 3

**Summary:**

The paper investigates the relationship between model size and domain specialization in the context of continued pretraining under compute-constrained scenarios. The authors aim to determine whether filtering training data for domain-specific content yields better performance than general-domain continued pretraining when computational resources are limited.
Key Contributions
Empirical Study on Domain Specialization vs. Model Scale:
The paper systematically evaluates how model size (1.5B, 3B, 7B, 14B parameters) interacts with domain-specialized continued pretraining across three domains: legal, medical, and accounting.
Controlled Training Setup:
All models undergo one epoch of continued pretraining on either:

A large, diverse web-based corpus (general),
Or filtered subsets focused on specific domains.
Evaluation via Domain-Specific Exams:
Performance is measured using standardized multiple-choice exams tailored to each domain (e.g., medical licensing-style questions), enabling precise assessment of knowledge retention and acquisition.
Discovery of a Power-Law Relationship:
Under compute constraints, specialized training follows a power-law improvement over general training.
Improved Compute Efficiency and Reduced Forgetting:
Specialized models achieve lower perplexity faster and exhibit less forgetting of previously acquired knowledge, suggesting more efficient use of compute during continued pretraining.

**Strengths:**

This paper presents a compelling and timely investigation into the interplay between model size and domain specialization during continued pretraining under compute-constrained settings. Its strengths span multiple dimensions—originality, quality, clarity, and significance—and collectively position it as a valuable contribution to the field of efficient language model adaptation.

1. Originality: High – Novel Problem Formulation with Fresh Insights
The paper’s originality lies in its novel framing of an understudied trade-off: how domain specialization interacts with model scale when compute is limited. While both scaling laws (e.g., Kaplan et al., Hoffmann et al.) and domain adaptation (e.g., Med-PaLM, LlamaLaw) have been widely studied, this work uniquely connects them through the lens of continued pretraining efficiency.
Key aspects of originality include:
Challenging a prevailing intuition: The hypothesis that larger models inherently benefit less from specialization due to greater capacity is intuitive but previously untested. The fact that the authors empirically disprove this assumption—showing instead that larger models gain more from specialization—is a striking and counterintuitive result.
Creative experimental design: By filtering a web-scale dataset using neural topic models to isolate legal, medical, and accounting domains, the authors construct a clean, controlled comparison between general and specialized data regimes—a methodologically sound approach rarely seen in prior specialization studies.
Power-law observation in specialization gains: Identifying a power-law relationship between model size and the relative benefit of domain filtering introduces a new quantitative lens for analyzing specialization strategies—an idea ripe for future theoretical exploration.
Thus, while none of the individual components (scaling laws, domain filtering, continued pretraining) are new, their synthesis into a coherent framework for evaluating data strategy under constraints represents a creative and original contribution.

2. Quality: Strong – Rigorous Methodology and Reproducible Design
The paper demonstrates high scientific rigor across several dimensions:
Controlled variables: Training all models for exactly one epoch on comparable data volumes ensures a fair evaluation under fixed compute budgets—an essential condition for meaningful conclusions about efficiency.
Model scale diversity: Including four distinct sizes (1.5B to 14B) enables robust trend analysis and supports claims about scalability.
Evaluation on standardized exams: Use of domain-specific multiple-choice benchmarks (e.g., analogous to USMLE or bar exams) provides objective, real-world-relevant metrics, avoiding proxy measures like perplexity alone.
Consistent patterns across domains: The observed trends hold across three disparate domains (legal, medical, accounting), suggesting the findings are not domain-specific artifacts but potentially generalizable principles.
Inclusion of forgetting analysis: Measuring retention of prior knowledge via perplexity on general text adds depth to the evaluation and strengthens claims about reduced catastrophic forgetting in specialized training.
While the paper does not release models or datasets (understandable given anonymity), the methodology is described clearly enough to allow replication by well-resourced labs.

3. Clarity: Excellent – Well-Structured and Accessible Presentation
Despite tackling a complex intersection of scaling, specialization, and transfer learning, the paper is exceptionally clear and logically structured:
Abstract and introduction set up the problem effectively, motivating both practical and theoretical interest in the question: "Should we filter data when continuing to pretrain?"
Hypothesis is explicitly stated, then cleanly refuted by evidence—this narrative arc enhances readability and impact.
Figures (e.g., Figure 1) appear designed to highlight key takeaways (power-law scaling advantage), though full visual access is limited in the blind submission format.
Technical terms (e.g., compute-equivalent replay, re-warming) are briefly contextualized, making the work accessible to a broad NLP audience.
Writing is concise and free of unnecessary jargon; logical flow moves naturally from motivation → hypothesis → experiment → results → implications.
Overall, the clarity significantly enhances the paper's persuasiveness and accessibility.

4. Significance: Broad and Practical Impact Across Communities
The significance of this work extends beyond a single application or architecture—it speaks to fundamental questions about how best to use finite compute resources in the era of large pretrained models.
Its impact can be felt across multiple communities:
Industry & Applied AI: Organizations fine-tuning LLMs for verticals (healthcare, law, finance) will find direct guidance: investing in high-quality, domain-filtered data may yield better returns than further scaling general data.
Efficient ML Research: The paper contributes to the growing body of work on "effective compute" — showing that smarter data selection can outperform brute-force scaling. This aligns with recent interests in data curation, weighting, and pruning (e.g., MinT, LLMLingua).
Scaling Law Theory: By demonstrating that data quality and relevance alter the optimal parameter–token ratio, the work suggests extensions to existing scaling laws to incorporate domain alignment factors—a promising direction for theory.
Sustainability & Equity: Reducing compute needs for high-performance specialized models lowers barriers to entry and reduces environmental costs, supporting more sustainable and inclusive AI development.

**Weaknesses:**

While the paper presents a compelling narrative with strong experimental design and significant implications, several weaknesses—though not fatal—limit the robustness, generalizability, and depth of its conclusions. Below is a detailed critique focused on specific shortcomings, supported by concrete suggestions for improvement.

1. Narrow Definition of "Specialization": Risk of Confounding Data Quality with Domain Focus
The core claim—that domain specialization improves performance under compute constraints—relies on comparing models trained on:
A broad web corpus (general),
Versus topic-filtered subsets (legal, medical, accounting).
However, filtering by topic may simultaneously improve data quality (e.g., removing low-signal text like social media posts or clickbait), which independently affects scaling laws (Tay et al., 2022; Hoffmann et al., 2022 showed data quality alters scaling constants).

2. Missing Analysis of Cross-Domain Generalization and Negative Transfer
The paper evaluates models only on in-domain exams, showing that specialized models outperform general ones. However, it omits evaluation on out-of-domain tasks, making it impossible to assess the trade-off between specialization gain and generality loss—a central concern in transfer learning.
For example:
Did the medical-specialized 14B model degrade on commonsense reasoning (e.g., HellaSwag)?
How much did legal specialization hurt performance on programming (e.g., HumanEval)?

**Questions:**

see above.

---

> ### Author Response · Authors · 2025-11-20
> **Response to Reviewer Tb5q**
>
> Thank you very much for your thoughtful review of our work. We appreciate your positive assessment of the paper’s originality and relevance. Our central goal is indeed not to restate classical scaling laws **[1, 2, 3]**, but to study how the scaling behavior of continued pretraining changes when we switch between two regimes under fixed compute: **(i)** general-domain continued pretraining and **(ii)** domain-specialized continued pretraining. To the best of our knowledge, this interaction between scaling behavior across these training regimes is unique to our results and methodology.
>
> Regarding the two main weaknesses you raised:
> > **1. Narrow Definition of “Specialization”: Risk of Confounding Data Quality with Domain Focus The core claim…**
>
> We agree that this is an important concern. If our domain-specialized datasets were simply higher-quality subsets of the general corpus, then the observed gains could be attributed only to data-quality improvements rather than to domain specialization. **However, given the results shown in Section 4.4, we find that this interpretation is inconsistent with our empirical results.** In this Section, we evaluate all models on a Portuguese general-domain test suite. If our results were obtained by a mere quality filter, we would expect the specialized models to also outperform the general models on this general-domain evaluation, since they would have been trained on "cleaner" data that should help overall language modeling.
>
> Instead, our results show the opposite: the general models (trained on unfiltered general Portuguese data) consistently achieve higher performance on the Portuguese general-domain test suite than both the specialized models and the original base models, across all model sizes, as shown in Figure 5 of Section 4.4.
>
> These findings are exactly what one expects from a domain specialization scenario: the specialized models are better on their own domain, but worse on out-of-domain tasks. Given our empirical results, we find that the specialized models are not behaving like “better general-purpose language models trained on higher-quality data.”
>
> These results are explained in more detail in Section 4.4 and Figure 5 of our study, especially in lines 434 to 445. Additionally, you can directly inspect the performance of both specialized and general models on this Portuguese general-domain test suite over the course of training in the Section B of our supplementary material, particularly in the middle row of Figures 6, 7, and 8. In these Figures, we can clearly see that the general models consistently improve more than the specialized models on the Portuguese general-domain test suite.
>
> > **2. Missing Analysis of Cross-Domain Generalization and Negative Transfer The paper evaluates models only on in-domain exams, showing that specialized models outperform general ones…**
>
> **We also disagree with this premise. Our paper does evaluate out-of-domain performance along two axes, both of which directly aim to represent the trade-off between specialization and general training in LMs, in terms of both performance and compute efficiency:**
>
> In Section 4.4, as already discussed previously in this response, we evaluate the resulting models on our general Portuguese test suite, which is out-of-domain for the legal, medical, and accounting specialized models. We observe that specialized models perform worse than the general models on this general-domain test suite, corroborating our interpretation that we are indeed studying a specialized training regime.
>
> In Section 4.3, we evaluate the models on the MMLU benchmark, using the high school and college partitions, which contain MCQA exams across diverse STEM and humanities domains. As a result of the trade-off between specialization and general training, we observe that both of our continued pretraining setups in Portuguese (specialized and general) exhibit forgetting of previously learned English knowledge, as both specialized and general models perform worse than the original base models. At the same time, as model size increases, specialized models become more sample efficient, and the performance gap between specialized and general models on MMLU shrinks, with the 14B specialized model approaching the general model’s performance.
>
> We see that this is a very interesting avenue that could be further expanded in future work. For example, a detailed comparison between our Portuguese legal-specialized model and the general Portuguese model on STEM and programming exams could reveal whether the general model maintains stronger performance in these areas, or whether domains closer to legal benefit more from specialization. **However, we view these investigations as complementary extensions of our work rather than core weaknesses of the current study.**
>
> ---
>
> ## References
>
> **[1]** Scaling laws for neural language models
>
> **[2]** Scaling laws for transfer
>
> **[3]** Training Compute-Optimal Large Language Models

---

### Official Review · Reviewer_tX4T · 2025-11-03

**Soundness:** 2
**Presentation:** 2
**Contribution:** 3
**Rating:** 2
**Confidence:** 3

**Summary:**

This paper investigates the relationship between domain specialization and model size in continued pretraining of Transformer LMs under compute-constrained scenarios. Using filtered datasets from ClueWeb 2022, the authors train Qwen2.5-based models (1.5B–14B) on both general and specialized (legal, medical, accounting) data.

**Strengths:**

Compute Efficiency and Forgetting Analysis:Introducing the SGER (Specialized-to-General Efficiency Ratio) metric
The Related Work section is comprehensive and well-situated in recent scaling law and domain-adaptation literature

**Weaknesses:**

Plots and cross-suite comparisons appear to use the test suite to pick the “minimum perplexity” checkpoint, then report those results. This is classic evaluation leakage; that minimum should be picked on a validation split that is disjoint from the reported test metrics


Power-law claims (and the SGER vs size trend) are fit on four points without confidence intervals, fit method details, or ablations. This risks over-interpreting noise. (You do acknowledge this in Limitations, but the paper still leans heavily on the regressions.) Consider reporting CI bands / bootstrapped SEs and fit diagnostics.

The paper follows Xia et al. and uses PPL on the correct answer letter; however, accuracy (and calibrated accuracy like log-prob of correct minus best distractor) is what many readers expect for MCQA.

**Questions:**

Did you use any data from the evaluation exams in the topic-classifier training or instruction prompts used for few-shot examples? Please clarify contamination controls.


How many random seeds per run?

---

> ### Author Response · Authors · 2025-11-20
> **Response to Reviewer tX4T Part [1/3]**
>
> Thank you for taking the time to review our work.
>
> We first address the main weakness raised by the reviewer.
>
> > **Plots and cross-suite comparisons appear to use the test suite to pick the “minimum perplexity” checkpoint, then report those results. This is classic evaluation leakage...**
>
> We believe this concern stems from a misunderstanding between standard practice in the scaling law literature and the evaluation conventions used for "benchmarking" a specific model. **In a scaling law study, it is common and intentional to summarize each training run by its best test loss under a fixed training recipe, and then study how that loss scales with model size, data, and/or compute.** Our methodology follows this convention and, in our view, does not undermine our conclusions.
>
> In **[1]**, the authors use cross-entropy loss on a held-out test partition of the WebText2 dataset (the same source dataset used for training) as the main metric for their scaling laws, as stated at the beginning of Section 2:
> > We record the loss on the WebText2 test distribution and on a selection of other text distributions.
>
> For the points used to fit their power laws, they explicitly perform early stopping based on this held-out test loss, effectively picking the minimum point for each run. Section 3.3 states:
> > We stopped training once the test loss ceased to decrease. We see that the resulting test losses can be fit with simple power-law
>
> > Thus for a given value of C we can scan over all models with various N to find the model with the best performance on step S
>
> **This is conceptually very close to what we do:** for each training run, we pick the checkpoint with the lowest loss on a held-out test evaluation dataset. The main difference is that, instead of cross-entropy on a slice of the pretraining corpus, we use PPL on the correct letter of our MCQA exams relevant for each specialization domain.
>
> In **[2]**, the authors go a step further and use the training loss as their primary metric for their scaling fits. Footnote 2 states that:
> > For simplicity, we perform our analysis on the smoothed training loss which is an unbiased estimate of the test loss, as we are in the infinite data regime (the number of training tokens is less than the number of tokens in the entire corpus).
>
> Thus, their fitted curves are an approximation to the "test" loss on the pretraining distribution. To fit their power laws, Section 3.1 describes:
> > Then, for each FLOP count, we determine which run achieves the lowest loss. Using these interpolants, we obtain a mapping from any FLOP count C, to the most efficient choice of model size N and number of training tokens D such that FLOPs(N, D) = C
>
> As a result, all of the selected points for their fit lie within the last 15% of checkpoints for each run; i.e., **they also deliberately select optimal and near-optimal checkpoints, thereby avoiding a "blind" choice of the checkpoint in regard to their "test" performance, which we understand that you are referencing as the "correct" approach.**
>
> In **[3]**, the authors similarly set aside 3% of their Python code dataset as a held-out test set, and for each run they train to convergence and select the optimal early stopping point on that held-out split. These optimal points are then used to fit their transfer scaling law, again following the same recipe as **[1]**.
>
> Multiple subsequent works adopt variants of this procedure. These studies are widely regarded as foundational in the literature, and their use of a held-out test evaluation distribution to select best checkpoints is treated as part of the definition of the training procedure whose scaling is being studied, not as "cheating" on a benchmark.
>
> In summary, when studying scaling laws, the goal is to characterize how a loss behaves as a function of model size, dataset size, and/or compute under a fixed training recipe. **The object of interest is the shape of these power laws, not the unbiased generalization of a single final checkpoint on a one-shot "blind" test.** Under this viewpoint, summarizing each run by its minimum loss on a fixed held-out test evaluation distribution is standard and appropriate. In our work, all models are trained under the same regime, and we apply the same selection rule consistently across sizes and domains. **This controlled setup is what supports our conclusions, ensuring a fair comparison specific to our controlled scenario.**
>
> ---
>
> ## References
>
> **[1]** Scaling Laws for Neural Language Models
>
> **[2]** Training Compute-Optimal Large Language Models
>
> **[3]** Scaling Laws for Transfer

---

> > ### Author Response · Authors · 2025-11-20
> > **Response to Reviewer tX4T Part [2/3]**
> >
> > > Power-law claims (and the SGER vs size trend) are fit on four points without confidence intervals, fit method details, or ablations. This risks over-interpreting noise. (You do acknowledge this in Limitations, but the paper still leans heavily on the regressions.) Consider reporting CI bands / bootstrapped SEs and fit diagnostics.
> >
> > Our main conclusions rely primarily on the legal models, where the log-log relationships exhibit a visually good fit to a power law (and, correspondingly, a log-linear relation for SGER), as specified in lines 348 to 351. We do not claim a universal scaling law that holds across all training scenarios. **In that sense, we believe the current methodology is sufficient to support our conclusions, given the scope and level of claims we make.**
> >
> > However, we agree that the presentation would benefit from clearer detail for fit diagnostics. In the revised version, we will include in the supplementary material confidence intervals and bootstrap-based standard errors for the main scaling plots.
> >
> > > The paper follows Xia et al. and uses PPL on the correct answer letter; however, accuracy (and calibrated accuracy like log-prob of correct minus best distractor) is what many readers expect for MCQA.
> >
> > We see this concern as closely related to the first one ("risk of evaluation leakage"). Our goal is not to optimize a particular model for these exams, but to study the relative scaling behavior of specialized versus general continued pretraining in a controlled setting. In this context, what matters is how performance scales with model size and compute across training regimes, rather than the exact absolute score of any single checkpoint. **PPL on the correct option is a natural choice for this purpose of scaling analysis, given its continuous nature compared to accuracy in MCQA, as argued by [4].** We view the reviewer’s preference for accuracy as more aligned with classic benchmarking work that aims to advocate for generalization of a specific model on a task, which is complementary but orthogonal to our primary objective here.
> >
> > **In summary, we disagree with the main weaknesses raised by the reviewer and do not believe they invalidate our results.** At the same time, we would welcome further discussion on the reviewer’s perspective regarding the validity of scaling law studies, particularly in relation to the notion of "evaluation leakage" raised here, regarding the goals of scaling studies and the practices adopted in the main cited works and related literature.
> >
> > ---
> >
> > ## References
> >
> > **[4]** Training Trajectories of Language Models Across Scales

---

> > > ### Author Response · Authors · 2025-11-21
> > > **Response to Reviewer tX4T Part [3/3]**
> > >
> > > Next, we address the questions raised by the reviewer:
> > >
> > > > **Did you use any data from the evaluation exams in the topic-classifier training or instruction prompts used for few-shot examples? Please clarify contamination controls.**
> > >
> > > **No. For training the topic classifier, we randomly sampled documents from ClueWeb2022, which consists of web pages.** These documents were then annotated with gpt-4o-2024-08-06 to obtain domain relevance labels for our target domains. Using this labeled dataset (document, relevance score), we trained BERT-based classifiers for each domain (legal, medical, and accounting). Further details are provided in Section 3.1, lines 205 to 222, and additional analysis of the relevance score distribution (Figure 9) and classifier training is included in Section C of the supplementary material.
> > >
> > > **The MCQA test suites used in this study are strictly held-out: they are never used in the topic classifier training, in the continued pretraining, or in any instruction prompts for few-shot examples.**
> > >
> > > Regarding potential contamination from ClueWeb2022 itself, the crawl date for ClueWeb2022 is late 2022. For our evaluation, **we only use exams that are administered after this crawl**, which mitigates the risk that exam questions appear in the pretraining data.
> > >
> > > > **How many random seeds per run?**
> > >
> > > **We use a single random seed per run.** As discussed in response to the first weakness, our focus is on the scaling behavior, rather than on precise point estimates for a specific checkpoint. In this regime, **additional seeds would primarily induce small perturbations in final PPL/Compute values and correspondingly minor shifts in the fitted curves, without changing the trends we report.** Supported by the existing literature on scaling laws, we note that prior work is consistent with our assumption: to the best of our knowledge, these studies do not report using multiple random seeds per run **[1, 2, 3, 4]**.
> > >
> > > ---
> > >
> > > ## References
> > >
> > > **[1]** Scaling Laws for Neural Language Models
> > >
> > > **[2]** Training Compute-Optimal Large Language Models
> > >
> > > **[3]** Scaling Laws for Transfer
> > >
> > > **[4]** Training Trajectories of Language Models Across Scales

---

### Meta-Review · Area_Chair_1Hbh · 2025-12-22

**Summary:**

This paper analyzes the effect of training models on domain-specific data versus general data by empirical analysis. Their key findings include that specialized models achieve better compute and performance efficiency than general training, particularly with larger models.

The paper received mixed reviews with multiple reviewers concerned about the clarity of writing, intended scope, and significance of the results. My own reading agrees with their assessments. I believe that although the paper explores an important problem, the limited analysis prevents determining sufficiently significant impact.

**Reviewer Concerns:**

- Clarity of writing, where the exact findings are unclear: I agree with the reviewer sentiment that the overall presentation of the paper is unclear. The rebuttal has not addressed this point.
- Small scale in evaluation and analysis: The rebuttal has clarified the evaluation metrics, but has not addressed the limited scale issue fully.
- Limited definition of specialization and confounding with just having a better dataset: The rebuttal has resolved this point.
- Limited analysis of cross-domain generalization: The rebuttal has partially clarified this point to show that specialized models perform poorly on out-of-distribution testing in one specific test setup, but this could be expanded further.
- Evaluation leakage: The rebuttal has clarified this point.
- Over-interpreting noise in power law: The authors acknowledge the concern as a limitation.

**Reviewer Scores:**

I do not believe the two reviewers with negative scores would increase their scores meaningfully as both reviewers seemed to have fundamental concerns with the paper. One reviewer’s core concern is clarity of the paper’s presentation, which has not been addressed, although some of the questions have been answered. All reviewers raise some questions about the scope of the paper and/or significance of effects.

---

### Decision · Program_Chairs · 2026-01-26

Reject